# NSCLC Cells Resistance to PI3K/mTOR Inhibitors Is Mediated by Delta-6 Fatty Acid Desaturase (FADS2)

**DOI:** 10.3390/cells11233719

**Published:** 2022-11-22

**Authors:** Marika Colombo, Federico Passarelli, Paola A. Corsetto, Angela M. Rizzo, Mirko Marabese, Giulia De Simone, Roberta Pastorelli, Massimo Broggini, Laura Brunelli, Elisa Caiola

**Affiliations:** 1Laboratory of Molecular Pharmacology, Department of Oncology, Istituto di Ricerche Farmacologiche Mario Negri IRCCS, 20156 Milan, Italy; 2Department of Pharmacological and Biomolecular Sciences, Università degli Studi di Milano, 20133 Milan, Italy; 3Protein and Metabolite Biomarkers Unit, Istituto di Ricerche Farmacologiche Mario Negri IRCCS, 20156 Milan, Italy

**Keywords:** non-small-cell lung cancer, BEZ235, drug resistance, PI3K/Akt/mTOR pathway, lipid metabolism

## Abstract

Hyperactivation of the phosphatidylinositol-3-kinase (PI3K) pathway is one of the most common events in human cancers. Several efforts have been made toward the identification of selective PI3K pathway inhibitors. However, the success of these molecules has been partially limited due to unexpected toxicities, the selection of potentially responsive patients, and intrinsic resistance to treatments. Metabolic alterations are intimately linked to drug resistance; altered metabolic pathways can help cancer cells adapt to continuous drug exposure and develop resistant phenotypes. Here we report the metabolic alterations underlying the non-small cell lung cancer (NSCLC) cell lines resistant to the usual PI3K-mTOR inhibitor BEZ235. In this study, we identified that an increased unsaturation degree of lipid species is associated with increased plasma membrane fluidity in cells with the resistant phenotype and that fatty acid desaturase FADS2 mediates the acquisition of chemoresistance. Therefore, new studies focused on reversing drug resistance based on membrane lipid modifications should consider the contribution of desaturase activity.

## 1. Introduction

The PI3K/Akt/mTOR pathway is an important signaling pathway regulating several essential cellular functions [1,2,3,4,5,6]. It is one of the most deregulated pathways in human cancer [6,7,8,9]. In particular, activating mutations of PI3K encoding genes (such as *PIK3CA*) or deletions in the negative regulators (such as PTEN) have been found in several human tumors, and this results in constitutive activation of the pathway, leading to aberrant growth, proliferation, and differentiation [10]. Several efforts were made to try to specifically target the major components of the pathway. Indeed, inhibitors of PI3K, Akt, and mTOR have been extensively studied at the preclinical level, and some have been used in clinical settings for in different tumor types [11,12,13]. At the preclinical level, very promising results were obtained with class-specific PI3K inhibitors, pan-PI3K inhibitors, Akt inhibitors, mTOR inhibitors, and dual PI3K-mTOR inhibitors, both in hematological and solid tumors [11,12]. However, in the clinic the benefits were below expectations and mainly restricted to a few tumor types [9,13,14,15,16,17]. At present, only two mTOR inhibitors (temsirolimus and everolimus) and three PI3K inhibitors (idelalisib, copanlisib, and alpelisib), both of which mainly target the delta isoform of class I PI3K, have been approved by the FDA and are on the market [18,19]. It should be noted that two of the three PI3K inhibitors have been approved only for the treatment of lymphomas.

The disappointing clinical results might be partly due to unexpected toxicities, a lack of biomarkers able to select patients potentially responsive to these therapies, and intrinsic resistance to treatments [20,21]. To overcome these challenges, researchers are still investigating the pathway and drugs that can inhibit it.

In this study, we explored the mechanisms associated with resistance to PI3K/Akt/mTOR inhibitors in non-small cell lung cancer (NSCLC) cells that were resistant to the usual PI3K/mTOR inhibitor BEZ235, underscoring biochemical pathways and proposing new interventions that may pave the way to overcome resistance. BEZ235 (also known as dactolisib) is a potent dual pan-class I (with a particular affinity for isoform α) PI3K/mTOR inhibitor belonging to the class of imidazoquinoline derivatives. It binds the ATP-binding pocket of the p110 subunit of PI3K and to the catalytic site of mTOR, so it is able to inhibit both mTORC complexes (mTORC1 and mTORC2), resulting in effective inhibition of tumor proliferation and growth [22,23,24]. In particular, we investigated changes in metabolic pathways in BEZ235 resistant versus sensitive cell lines, with mTOR playing a pivotal role in cellular metabolism control [25,26,27,28]. It is well established that KRAS-mutated NSCLC cells display a distinct metabolic profile [29,30] and that *KRAS* mutations lead to the activation of the MAPK and PI3K/Akt7/mTOR pathways [31]. Thus, we also evaluated the metabolic changes considering the G12C *KRAS* mutational status.

## 2. Methods

### 2.1. Generation of BEZ235 Resistant Clones and Drug Treatments

NSCLC H1299 cells overexpressing wild-type (wt) or G12C mutant KRAS [29,30,32,33] were used to generate BEZ235-resistant clones.

Wt or KRAS G12C mutant H1299 cells were initially treated with 25 nM BEZ235, and the surviving cells were re-plated and re-treated with increasing concentrations of the drug. The BEZ235 concentration was raised to 500 nM, and after 40 and 38 passages (for H1299 KRAS wt and H1299 G12C, respectively), cells were frozen and maintained in culture without further drug treatment.

The activity of BEZ235 was evaluated using the MTS test. Cells (sensitive and resistant) were seeded in 96-well plates at the desired concentrations. Twenty-four hours after seeded cells were treated with increasing concentrations of BEZ235 and maintained at 37 °C for a further 72 h. MTS was added (10 µL/well) and absorbance at 490 nm was measured using the Glo Max plate reader (Promega, Madison, WI, USA). For each concentration, the experiment was repeated for a total of 6 replicates, and at fixed time intervals (about 3–4 weeks) BEZ235 activity was tested to verify that the resistant clones maintained their behavior.

The response of sensitive and resistant clones to other drugs was evaluated using the same conditions as for BEZ235. The drugs analyzed were the pan-PI3K inhibitor BKM120, the isoform-specific PI3K inhibitors PIK75, TGX22, and CAL101, the mTOR inhibitors Torin-1 and Ink128, the dual-PI3K/mTOR inhibitor PF05212384, the Akt inhibitor ARQ-751, and the DNA-damaging agent doxorubicin. The combination experiments with BEZ235 and betullin (a sterol regulatory element-binding protein (SREBP) inhibitor) were performed by treating the cells with increasing concentrations of BEZ235 after maintaining the cells for seven days in culture with betullin, or by co-treating the cells with the two drugs. All the compounds used were purchased from TargetMol (Wellesley Hills, MA, USA).

### 2.2. Western Blot Analyses

Proteins were extracted and visualized as reported in [34]. Separation of nuclear protein extracts from cytoplasmic proteins was performed by using a nuclear extraction kit (AbCam, Cambridge, UK) following manufacturer’s instructions. Immunoblotting was carried out with the following antibodies: anti-p70S6K(Thr389) #9206, anti-p70S6K #9202, anti-pAkt (Ser473) #4060, anti-Akt #9272, anti-S6(Ser235/236) ribosomal protein #2211, anti-S6 ribosomal protein 2217#, anti-4E-BP1(Thr37/46) #2855, anti-4E-BP1 #9644, anti-Acetyl-CoA carboxylase (ACC) #3662, anti- Stearoyl-CoA desaturase 1 (SCD1) #2794, and anti-Fatty Acid Synthase (FASN) #3189, provided by Cell Signalling Technology (Danvers, MA, USA). Anti-Erk #sc94, anti-Erk (Tyr204) #sc7383, anti-SREBP1 #sc-365513, anti-actin #sc-47778, and anti-lamin B #sc-374015 were obtained from Santa Cruz Biotechnology (Dallas, TX, USA).

### 2.3. Real-Time RT PCR

Total RNA was extracted and reverse transcribed as reported in [29]. The mixture was amplified by a 7900HT Sequence Detection System (Thermo Fisher Scientific, Waltham, MA USA). Actin was used as the internal control. Primers and TaqMan probes were purchased for all genes as ready-to-use solutions (Assay on Demand, Thermo Fisher Scientific, Waltham, MA USA). mRNA relative expression levels were calculated with the 2-ΔΔCt method. Wt for wt_bez and G12C for G12C_bez clones were set to 1. Two samples that showed at least a 2-fold difference in expression were considered differently expressed.

### 2.4. Metabolomics Sample Preparation

H1299 cell clones were grown for 48 h in biological triplicate. At 48 h, all four clones were assumed to have the same proliferation rate. All of the cell lines (three biological replicates/clone) were rapidly rinsed in a saline solution quenched with liquid nitrogen and stored at −80 °C until the analysis. Extraction was done as described in [29].

### 2.5. Untargeted Metabolomics

Flow injection analysis high resolution mass spectrometry (FIA-HRMS) was used for untargeted metabolomics of H1299 cell clones as previously described [35]. All data were processed and analyzed in Matlab R2016a (The Mathworks, Natick, MA, USA) using an in-house developed script [35].

### 2.6. Targeted Metabolomics

A targeted quantitative approach using a combined direct-flow injection and liquid chromatography tandem MS/MS assay (AbsoluteIDQ^®^ p180 kit, Biocrates, Innsbruck, Austria) was applied, as previously published [30].

### 2.7. Metabolic Pathway Analysis

For biological interpretation of the metabolite dataset, we mapped the significant metabolites derived from both untargeted and targeted approaches into a biochemical network using MetaboAnalyst (www.metaboanalyst.ca, accessed on 15 June 2022). Enrichment analysis (EA) tools were used to identify metabolic pathways most likely to be associated with the BEZ235 acquired resistance.

### 2.8. Free Fatty Acids (FFA)

FFAs (palmitic and oleic acids) were measured in H1299 cells clones grown in culture for 48 h (1 × 106 cells) as detailed in [34].

### 2.9. Free Cholesterol

Intracellular free cholesterol levels were determined using a 1200 Series HPLC system (Agilent Technologies, Santa Clara, CA, USA) interfaced to an API 5500 triple quadrupole mass spectrometer (Sciex, Thornhill, ON, Canada) as described in [36].

### 2.10. Lipid Droplets Detection

Cells were seeded at 15,000 cells/mL on glass coverslips. After 48 h, cells were fixed with 4% paraformaldehyde, and lipid droplets were detected by the Lipid Droplets Fluorescence Assay Kit (CaymanChem, AnnArbour, MI, USA), following the manufacturer’s instructions. Briefly, the kit is based on Nile Red staining, which is a fluorescent dye that is very sensitive and specific for neutral lipids, which constitute lipid droplets [37]. Then, nuclei were counterstained with DAPI. Fluorescence was visualized at the confocal microscope (IX81, Olympus, Shinjuku, Japan).

### 2.11. Membrane Fluidity Assay

Cells were seeded in black 96-well plates at a concentration of 50000 cells/mL and after 48 h membrane fluidity was determined with a membrane fluidity kit (AbCam, Cambridge, UK) following the manufacturer’s instructions. The assay exploits a peculiar property of pyrenedecanoic acid (PDA), which, when added to cells, incorporates into the cell’s plasma membrane and forms homodimers, with the rate of formation proportional to the membrane fluidity. When dimers form, the emission spectrum of the pyrene probe shifts dramatically from 400 nm to 470 nm. Briefly, PDA was added to the cells, and they were incubated for 1 h at 25 °C in the dark. After unincorporated PDA was removed from the samples, fluorescence was read at 400 nm (monomer) and 470 nm (dimers) with the GloMax plate reader (Promega, Madison, WI, USA). The normalized excimer to monomer fluorescence ratio was then calculated, as a quantitative monitoring of the membrane fluidity.

### 2.12. Statistical Analyses

Statistical analyses were performed using GraphPad Prism V8 (San Diego, CA, USA). The specific tests used in the experiments are reported in the figure legends. A *p*-value < 0.05 was considered statistically significant.

## 3. Results and Discussion

### 3.1. mTOR, Rather Than PI3K- Related Pathways, Are Mainly Involved in the Resistance to BEZ235

To investigate new mechanisms of acquired resistance to PI3K/Akt/mTOR axis inhibition, we generated BEZ235-resistant clones starting from our NSCLC H1299-derived clones, expressing wt or G12C-mutated KRAS [29,30,32,33]. Wt and G12C cells are similarly responsive to the dual PI3K and mTOR inhibitor BEZ235, with an IC50 of approximately 40-50 nM for both clones. For the induction of resistance, the cells were initially treated with 25 nM BEZ235, and the surviving cells were re-plated and retreated with the same concentration of the drug. After 2 passages, the concentration of BEZ235 was increased to 50 nM, then to 100 nM (after 6 passages) to arrive at 500 nM after 18 passages. This concentration was maintained for 20 additional passages. Figure 1A reports the growth inhibitory activity of BEZ235 in parental wt and G12C cells and in resistant clones, wt_bez and G12C_bez, obtained after 40 and 38 passages in vitro with increasing concentrations of BEZ235, respectively. The activity of BEZ235 in resistant cells was determined after at least one further passage in vitro without the drug. The activity of BEZ235 was lower in both wt_bez and G12C_bez cells compared to their corresponding parental cells. For the wt cells, the IC50 of BEZ235 was 52.59 nM and became 282.4 nM in the wt_bez cells (RI = 5.4). In the G12C cells, BEZ235 had an IC50 of 68.14 nM and in the G12C_bez cells of 398.5 nM (RI = 5.8).

We first checked for the presence of overexpression of the multidrug resistance (MDR-1) and ATP-binding cassette, sub-family G, isoform 2 (ABCG2) genes, both known to be involved in cell efflux of several drugs [38,39,40], by measuring mRNA expression. Figure 1B,C clearly show that BEZ235-resistant clones do express, at the mRNA level or even lower (in the case of MDR-1), the two genes compared to their respective parental cells, thus excluding increased expression of *MDR-1* and *ABCG2* as causes for the resistance. The similar viability of the four clones after treatment with increasing concentrations of doxorubicin, known to be one of the most studied targets of MDR-1 and ABCG2 [41], confirmed that these extrusion pumps were not involved in the resistance (Appendix A).

We then decided to analyze at molecular level the expression and phosphorylation of proteins belonging to the pathway targeted by BEZ235. Figure 1D reports a representative western blot, obtained from cell extracts taken from exponentially growing cells. Western blot analysis shows a greater phosphorylation of the KRAS downstream protein ERK in the mutated KRAS cell lines, G12C and G12C_bez. This increased phosphorylation is a consequence of the constitutive activation of the KRAS protein because of the presence of the G12C mutation in both clones. As far as proteins belonging to PI3K/mTOR pathways, significant alterations were observed in the activation of Akt protein, downstream PI3K, and p70, a direct target of mTOR, being higher in the BEZ235 resistant clones. On the other hand, the activation levels of the proteins S6, activated by p70, and 4EBP1, directly inhibited by mTOR, were not different. We therefore hypothesize that the pathways aimed at activating protein synthesis (those of S6 and 4EBP1; [42]) are not the main players in BEZ235 resistance mechanisms and that other pathways, probably regulated by p70S6K or Akt, might be involved.

Being BEZ235 a dual PI3K/mTOR inhibitor, we tested whether a collateral resistance was observable using drugs acting at different levels in the PI3K/Akt/mTOR pathway, as well as which step of the axis was most involved in the resistance. Figure 1E–L show the dose-response curves obtained from the cytotoxicity experiments performed on the clones treated with increasing concentrations of the drugs. Both resistant clones were equally sensitive (compared to the parental cells) to the pan-PI3K inhibitor BKM120 (Figure 1E) and to the alpha isoform-specific PI3K inhibitors (Figure 1F). In the case of treatment with CAL-101 (Figure 1G), an inhibitor of PI3Kδ, all the clones were resistant. Treatment with TGX-221 (Figure 1H), an inhibitor of PI3Kβ, induced a slight increase in resistance in resistant BEZ235 cell lines compared to the sensitive ones. The clones were also equally sensitive to the pan-Akt inhibitor ARQ751 (Figure 1I) and all insensitive to the mTORC1 inhibitor rapamycin (Figure 1J), but were cross-resistant to the mTORC1 and mTORC2 inhibitors torin-1 and INK-128 (Figure 1K,L). Collectively, our data suggest that the resistance is more related to mTOR than PI3K.

### 3.2. BEZ235 Resistance Alters the Membrane Lipid Compartment

mTOR has a role in controlling cell metabolism (e.g., regulation of glycolysis, pentose phosphate pathway, and de novo lipogenesis) [43]. Furthermore, considering our previous findings about metabolic changes associated with the presence of a mutated KRAS in isogenic cells [29,30,32], we used both targeted and untargeted metabolomics approaches to profile the metabolic asset of BEZ235 resistant clones in comparison to their respective parental cell lines. We identified the significant alteration of 164 and 133 metabolites in wt_bez and G12C_bez resistant cells, respectively, relative to their sensitive counterparts (Appendix A). Pathway enrichment analysis highlighted the deregulation of metabolites mainly involved in the urea cycle and ammonia recycling pathways (Appendix A). This is consistent with the knowledge that changes in cellular metabolism not only result in tumor progression but also contribute to cancer cells’ resistance to drug treatments [44]. Indeed, cancer cells may modulate glutamine metabolism and its interconnected ammonia recycling and urea cycle pathways, driving drug resistance [45,46,47].

Among the significantly different metabolites, we observed a strikingly higher proportion of lipid species in both resistant clones (105 out of 164 in wt_bez and 92 out of 133 in G12C_bez) relative to the sensitive ones (Figure 2A,B). The untargeted metabolomics approach indicated a widespread increment of phosphatidylethanolamines, ceramides, mono- and triglyceride species in both wt_bez and G12C resistant clones (Appendix A). Focusing on the quantified lipid species in both resistant cells, we confirmed the increased levels of lysoPCs, PCs, SMs, and in their saturated (SFA), monounsaturated (MUFA), and polyunsaturated (PUFA) forms with no distinction in length (number of carbons from 16 to 42) and no relation to the KRAS mutational status (Figure 2C,D and Appendix A).

In addition to central cellular metabolic adaptation in cancer resistance, evidence shows that lipid metabolism is also involved in developing cancer resistance [48]. Since we found that BEZ235-resistant cells had a higher abundance of lipid species relative to sensitive ones, we further investigated whether such an increase was due to either the storage and/or biosynthesis of these species in the cells. We first analyzed the amount of lipid droplets in sensitive and resistant clones. Lipid droplets are intracellular organelles present in nearly all cells. Their main function is to store oil-based reserves of metabolic energy and components of membrane lipids, which are crucial for survival [49]. Using the Nile Red staining procedure, we did not find differences in lipid droplets between sensitive and resistant cells. In general, G12C cells (both sensitive and resistant) showed lower amounts of lipid droplets compared to wt KRAS expressing cells (Figure 2E). Therefore, we can exclude the association between lipid droplets and BEZ235 resistance.

Several studies suggest that altered lipid composition and physical properties of cell membranes contribute to the chemoresistance of cancers [50]. The most common lipid alterations in drug-resistant cancer cells are the increased abundance of phospholipids, sphingolipids, and cholesterol content and the modulation of the ratio between saturated/unsaturated lipid content [50,51]. In our setting, we did not observe alterations in the level of free cholesterol between sensitive and resistant clones. Free cholesterol is primarily incorporated in the plasma membrane and is essential for the membranes’ physical integrity; indeed, its membrane enrichment up to 20–40% results in a more rigid membrane bilayer [52] (Figure 2F). Saturation status of lipids is also involved in membrane packing and fluidity; indeed, modification of the saturated/unsaturated lipid ratio results in an alteration of lipid packing in the membrane and consequently modulates membrane fluidity, which is a crucial phenomenon in cancer drug resistance [44,51]. In agreement with the increased amount of MUFA and PUFA, we found that both BEZ235-resistant cells displayed a statistically significant increase in plasma membrane fluidity compared to parental cells (Figure 2G).

### 3.3. BEZ235 Resistance Is Linked to Increased Desaturase Expression and Activity

We then analyzed the involvement of SREBP1 in de novo lipogenesis and eventually in the acquisition of BEZ235 resistance in our models. SREBP-1 protein is a transcription factor master regulator of lipogenesis [53,54], which acts by activating the expression of more than 30 genes involved in the synthesis and uptake of cholesterol, fatty acids, triglycerides, and phospholipids [55,56,57]. The aberrant activation of this pathway leads to the synthesis of fatty acids, which function as essential building blocks of membranes. SREBP-1 is synthesized as an inactive protein that is activated upon proteolytic cleavage prior to nuclear translocation [58]. Furthermore, the nuclear accumulation of active SREBP-1 requires the activity of the mTORC1 complex [59]. We performed western blot analysis to assess SREBP-1 protein levels on both total (Figure 3A) and nuclear (Figure 3B) extracts from sensitive and resistant cells, but we did not observe any difference that could explain its role in BEZ235 resistance. Treatments with different schedules (co-treatment, pre-treatment for seven days) and concentrations (ranging from 1 μM to 10 μM, data not shown) of betullin, a SREBP1 inhibitor, were in fact not able to restore the sensitivity to BEZ235 both in wt_bez and G12C_bez clones. As such, it is unlikely that SREBP-1 might have a role in driving resistance in our model (Appendix A). However, in mammalian cells, three different SREBP isoforms, SREBP-1a, SREBP-1c, and SREBP-2, have been identified, having only partially overlapping roles in the regulation of fatty acid and cholesterol synthesis [60,61]. In particular, SREBP-1c is mainly involved in modulating fatty acid synthesis and de novo lipogenesis, while SREBP-1a takes part in two pathways, SREBP-1c and SREBP-2, specific to cholesterol metabolism [62]. Therefore, we will analyze the different isoforms in future experiments to finally exclude the involvement of SREBP1 (and in particular SREBP1-c) in the resistance observed.

We also examined whether increased lipid abundance in resistant clones was due to changes in the key players of de novo lipogenesis by monitoring the expression and protein abundance of the lipogenic enzymes Acetyl-CoA carboxylase isoforms 1 and 2 (ACC1, ACC2, coded by the *ACACA* and *ACACB* genes, respectively), Fatty Acid Synthase (FASN), and Stearoyl-CoA desaturase 1 (SCD1) [63]. As reported in Figure 3C, we did not assist to an increase in their protein levels in BEZ235 resistant clones, but in G12C_bez cells rather to a decrease in the expression of SCD1. Accordingly, real-time PCR confirmed the absence of any increased expression of these enzymes. There was actually a trend toward a reduction in the expression of all the enzymes in BEZ235 clones compared to their parental counterparts (Appendix A). Consistent with the absence of alteration in de novo lipogenesis, the BEZ235 resistant clones had comparable levels of the main products of this pathway [64,65], the palmitic (product of FASN) and oleic (product of SCD1) free fatty acids (Figure 3D).

The results obtained so far on the regulators, enzymes, and products related to the de novo lipogenesis process do not explain the increase in MUFA and PUFA observed in the clones that are stably resistant to BEZ235.

The phospholipid fatty acid chain length and the number and position of double bonds can markedly influence membrane fluidity, permeability, and stability [66]. Consequently, we evaluated the contribution of fatty acid desaturation and chain elongation in exerting the lipid modifications observed in our resistant clones by measuring the expression of fatty acid elongases 2 and 5 (ELOVL2 and ELOVL5) and fatty acid desaturases 1 and 2 (FADS1 e FADS2), responsible for the elongation and desaturation of fatty acid lipid chains [67,68].

The analysis showed similar expression levels of the elongases ELOVL2 and ELOVL5 and the desaturase FADS1 in all clones (Figure 3E,F). Instead, a significant overexpression of the FADS2 enzyme is observed in both BEZ235-resistant clones compared to sensitive ones (Figure 3F). Accordingly, the ratio of MUFA/SFA and PUFA/SFA, an index of desaturase activity, was statistically significantly increased in BEZ235-resistant cells (Figure 3G).

FADS2 mainly plays a role in desaturation by introducing a double bond at the Δ6 position of the fatty acid chain, and this is the first rate-limiting enzyme for the conversion of upstream fatty acids into PUFAs [69,70]. PUFAs are critical for the regulation of membrane biophysical properties and the functions of transmembrane proteins and peripheral membrane proteins [59,65]. In accordance with our observations, the increased FADS2 expression is likely linked to the increased PUFA levels and membrane fluidity observed in the BEZ235 resistant clones.

## 4. Conclusions

The PI3K pathway is the most frequently activated pathway in human cancers. Consequently, several compounds targeting the various nodes of this pathway have been developed. However, the majority of these compounds have been unsuccessful in patients due to high levels of toxicity as well as their inability to downregulate the pathway, which is partially mediated by the intrinsic adaptive response of tumors. The PI3K pathway is a key pathway regulating cell metabolism. Therefore, we focused on the characterization of the metabolic changes underpinning BEZ235 resistance in NSCLC cells. The observed general central cellular metabolic adaptation in BEZ235 resistant clones comes as no surprise, but we interestingly identified the central involvement of lipid modulation, in which the key contribution of FADS2-mediated fatty acid desaturation in increasing membrane fluidity is the peculiar feature of our BEZ235 resistance model. Understanding the role of the plasma membrane is pivotal to overcoming drug resistance. The interplay between the membrane fluidity and the efficacy of anticancer drugs is not restricted to the function of the membrane as a barrier to drug penetration but includes more complex mechanisms that the cell implements to maintain the homeostasis of the membrane’s physics to adapt to continued anticancer drug exposure. The precise molecular mechanisms linking BEZ235 resistance to the FADS2 enzyme, as well as how interfering with desaturase activity and possibly membrane fluidity may counteract BEZ235-acquired resistance, remain unknown.

## Figures and Tables

**Figure 1 cells-11-03719-f001:**
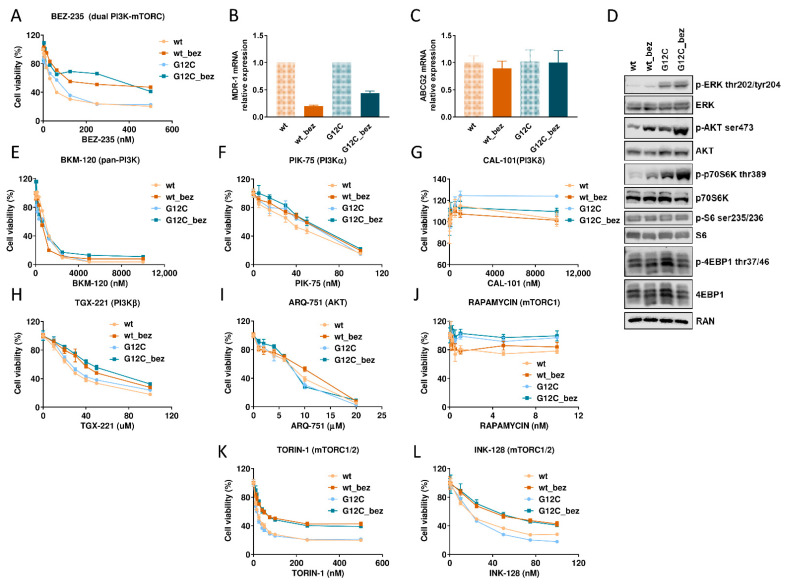
(**A**) Dose-response curve of the clones treated with BEZ235 at different concentrations. The means and standard deviation (SD) of at least three independent experiments are plotted. Statistical analysis was carried out through the two-way ANOVA test and the Bonferroni post hoc test for multiple comparisons and is reported in Appendix A. (**B**) Histograms representing the relative expression of *MDR-1* mRNA levels. Actin was used as a housekeeping gene for normalization. The means of three replicates and SD are reported. (**C**) Histograms representing the relative expression of *ABCG2* mRNA levels. Actin was used as a housekeeping gene for normalization. The means of three replicates and SD are reported. (**D**) Western blot analysis of MAPK and PI3K/mTOR pathway proteins in wt, wt_bez, G12C and G12C_bez clones at basal conditions. Ran was used as a loading control. (**E**–**L**) Dose-response curves of the clones treated with PI3K/Akt/mTOR inhibitors at increasing concentrations. On each graph, the specific drug with the corresponding target is reported. The means and standard deviation (SD) of at least three independent experiments are plotted.

**Figure 2 cells-11-03719-f002:**
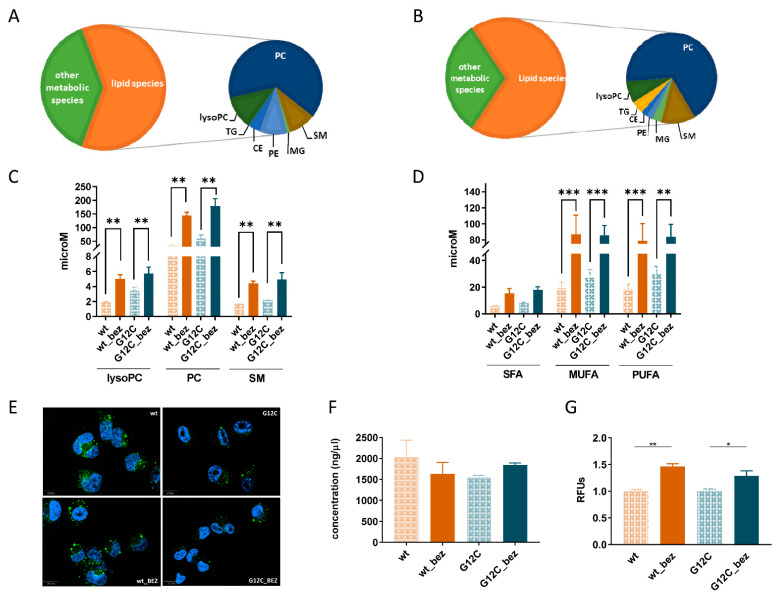
(**A**) Pie chart summarizing the metabolic class distribution of the statistically significant metabolites between the wt and wt_bez clones. (**B**) Pie chart summarizing the metabolic class distribution of the statistically significant metabolites between G12C and G12C_bez clones. (**C**) Sum of intracellular concentrations of lysophosphatidylcholines (lysoPC), phosphatidylcholines (PC), and sphingomyelins (SM) in wt, wt_bez, G12C, and G12C_bez clones. ** indicates statistically significant differences with *p* < 0.01 (two-way ANOVA and Tukey Kramer post-hoc test). (**D**) Sum and distribution of intracellular concentrations of saturated (SFA), monounsaturated (MUFA), and polyunsaturated (PUFA) lipid species. ** and *** indicate statistically significant differences with *p* < 0.01 and *p* < 0.001, respectively (two-way ANOVA and Tukey Kramer post-hoc test). (**E**) Representative images of lipid droplets detected by Nile Red fluorescent dye (green) in all the clones. DAPI (blue) was used to counterstain nuclei. The scale bar is 20.0 µM. (**F**) Intracellular concentration of free cholesterol. (**G**) Membrane fluidity expressed as relative fluorescence units (RFUs) of the ratio between PDA dimer and monomers, detected by a membrane fluidity kit (Abcam, see Section 2). * and ** indicate statistically significant differences with *p* < 0.05 and *p* < 0.01, respectively (one-way ANOVA and Bonferroni post-hoc test). LysoPC, Lysophosphatidylcholine, PC, Phosphatidylcholine, PE, lysophosphatidylethanolamine, CE, Ceramide, SM, Sphingomyelin, MG, monoacylglyceride, and TG, Triglyceride.

**Figure 3 cells-11-03719-f003:**
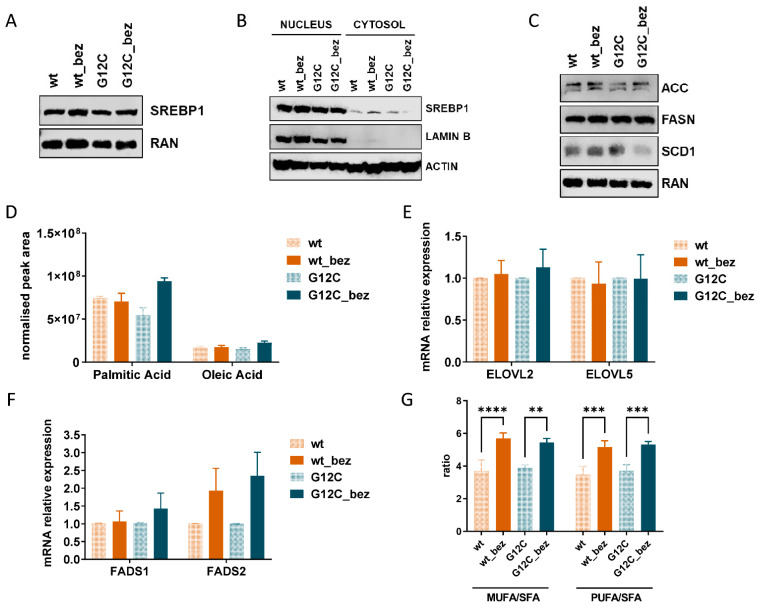
(**A**) Western blot analysis of SREBP1 protein expression in wt, wt_bez, G12C, and G12C_bez clones at basal conditions. Ran was used as a loading control. (**B**) Western blot analysis of SREBP1 protein expression in nuclear (left) and cytosolic (right) extracts of wt, wt_bez, G12C, and G12C_bez clones at basal conditions. Actin was used as a loading control. Lamin B was used as a nuclear extract marker. (**C**) Western blot analysis of ACC, FASN, and SCD1 protein expression in wt, wt_bez, G12C, and G12C_bez clones at basal conditions. Ran was used as a loading control. (**D**) Intracellular levels of Palmitic and Oleic acid in wt, wt_bez, G12C, and G12C_bez clones. (**E**) Histograms representing relative mRNA expression of the *ELOVL2* and *ELOVL5* genes. Actin was used as a housekeeping gene for normalization. The means of three replicates and SD are reported. (**F**) Histograms representing relative mRNA expression of the *FADS1* and *FADS2* genes. Actin was used as a housekeeping gene for normalization. The means of three replicates and SD are reported. (**G**) Estimated desaturase activity in the clones based on MUFA/SFA and PUFA/SFA ratio. **, ***, and **** highlight statistically significant differences with a *p* < 0.01, *p* < 0.001, and *p* < 0.0001, respectively (one-way ANOVA and Tukey Kramer post-hoc test).

## Data Availability

Not applicable.

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
