# Peer review of "NSCLC Cells Resistance to PI3K/mTOR Inhibitors Is Mediated by Delta-6 Fatty Acid Desaturase (FADS2)"

_cells, 2022, doi:10.3390/cells11233719_

Round 1

Reviewer 1 Report

Constitutive activation of the signaling pathways in cancer cells leads to aberrant growth, proliferation and differentiation. Molecules of signaling pathways are often used as drug targets.  Thus, detailed information about signaling cascades is critical for future drug discovery.

minor comments:

1.      The font size on histograms in Fig1, Fig 2G, Fig 3D-F is too small, please, enlarge it.  

2.      “Both resistant cells were equally sensitive (compared to the parental cells) to the pan PI3K inhibitor BKM120 (Figure 1E) and 258 to the isoform specific (alpha and delta) PI3K inhibitors (Figures 1F and 1G)”.  It is more correct to say that the clones were insensitive to inhibitors in Fig1G, L.

3.      Line 257: “Both resistant cells…” to change for “Both resistant cell clones…”

4.      Fig 2A, B add abbreviations for lipids in the figure legend. The pieces of pies should be signed more clearly.

5.      Line 343: RLUs should be changed for RFUs

Author Response

Point 1. The font size on histograms in Fig1, Fig 2G, Fig 3D-F is too small, please, enlarge it.

Response1. Following the reviewer's suggestion, we improved the font of the histograms.

Point 2. “Both resistant cells were equally sensitive (compared to the parental cells) to the pan PI3K inhibitor BKM120 (Figure 1E) and 258 to the isoform specific (alpha and delta) PI3K inhibitors (Figures 1F and 1G)”. It is more correct to say that the clones were insensitive to inhibitors in Fig1G, L.

Response 2. In accordance with the reviewer, we modified the manuscript text accordingly.

Point 3. Line 257: “Both resistant cells…” to change for “Both resistant cell clones…”

Response 3. In accordance with the reviewer, we modified the manuscript text accordingly.

Point 4. Fig 2A, B add abbreviations for lipids in the figure legend. The pieces of pies should be signed more clearly.

Response 4. In accordance with the reviewer, we improved the pie figure (2 A, B) and inserted the lipid abbreviations in the figure legend. 

Point 5. Line 343: RLUs should be changed for RFUs.

Response 5. Changed accordingly in the text.

Reviewer 2 Report

This is an efficient study aiming to find molecular alterations of these resistance mechanisms by developing resistance to BEZ235, a PI3K-mTOR inhibitor, which is a human non-small cell lung cancer cell line and conducted in two different lines, with and without KRASG12C mutation is seen. The use of both RAS mutant and RAS wild cell lines makes the study more meaningful. I like the way you present. Congrats.

Author Response

We thank the reviewer two for his/her appreciation of our manuscript.

Reviewer 3 Report

In this manuscript, the authors have investigated the possible mechanism of NSCLC cells resistant to BEZ235. The authors generated resistant NSCLC cell lines (WT and KRAS mutation) to BEZ235 and measured their metabolic profiles. By comparison, the authors indicated that the lipid alteration was associated with BEZ235 resistance. Overall, their conclusions suggested that the plasma membrane played an important role in drug resistance and complicated mechanisms were involved in drug treatment. However, there are some questions still needed to be answered:

1) Could the authors briefly describe BEZ235 in the introduction? In addition, could the authors explain why they chose BEZ235 to generate resistant cell lines instead of current FDA-approved drugs (idelalisib, copanlisib, and alpelisib)?

2)     As described by the authors, BEZ235 was considered as a dual PI3K/mTOR inhibitor. Could the authors briefly explain why BEZ235 resistant cell lines show resistance to other mTOR inhibitors, not other PI3K inhibitors? Does PI3K inhibitors have different mechanism to inhibit the activity of PI3K while mTOR inhibitors have similar mechanism?   

3)      Line 75, 490nm, not 490nM.

Author Response

Point 1. Could the authors briefly describe BEZ235 in the introduction?

Response 1. In accordance with the reviewer, we added some information about BEZ235 in the introduction.

Point 2. In addition, could the authors explain why they chose BEZ235 to generate resistant cell lines instead of current FDA-approved drugs (idelalisib, copanlisib, and alpelisib)?

Response 2. We thank the reviewer for the concern raised. We agree that other compounds could be used to generate resistant cell lines. The drugs cited by the reviewer were PI3K-only and isoform-specific inhibitors, while we wanted to explore the mechanisms of resistance after mTOR inhibition, too. The simultaneous inhibition of the two kinases (and of the downstream effectors of both) was found to be more effective than the single inhibition, so we thought that investigating the mechanisms of resistance after the combined inhibition of PI3K and mTOR would be more interesting. In addition, we have not started generating resistant clones in a recent time (in fact several passages were required to obtain the resistance). If so, we would have probably chosen another compound.

Point 3. As described by the authors, BEZ235 was considered as a dual PI3K/mTOR inhibitor. Could the authors briefly explain why BEZ235 resistant cell lines show resistance to other mTOR inhibitors, not other PI3K inhibitors? Does PI3K inhibitors have different mechanism to inhibit the activity of PI3K while mTOR inhibitors have similar mechanism?

Response 3. We thank the reviewer for the concern raised. The hypothesis suggested by the reviewer could be the explanation of the behaviour observed. We used both pan-PI3K and isoform-specific inhibitors, which have different affinities for the different isoforms. Apart from rapamycin, able to inhibit the mTORC1 complex, the two mTORCs inhibitors have instead a very similar mechanism.

Point 4. Line 75, 490nm, not 490nM.

Response 4. In accordance with the reviewer, we modified the manuscript text accordingly.